# Effects of IFIH1 rs1990760 variants on systemic inflammation and outcome in critically ill COVID-19 patients in an observational translational study

Laura Amado-Rodríguez[1,2,3,4†], Estefania Salgado del Riego[5†],
Juan Gomez de Ona[3,6,7], Inés López Alonso[2,3,4], Helena Gil-Pena[3,8],
Cecilia López-Martínez[2,3,4], Paula Martín-Vicente[2,3,4], Antonio Lopez-Vazquez[3,6,7],
Adrian Gonzalez Lopez[2,9], Elias Cuesta-Llavona[3,10], Raquel Rodriguez-Garcia[1,2,3],
Jose Antonio Boga[3,11], Marta Elena alvarez-Arguelles[3,11],
Juan Mayordomo-Colunga[2,3,8,10], Jose Ramon Vidal-Castineira[3], Irene Crespo[2,3,12],
Margarita Fernandez[4], Loreto Criado[12], Victoria Salvadores[12],
Francisco Jose Jimeno-Demuth[13], Lluis Blanch[2,14], Belen Prieto[12,15],
Alejandra Fernandez-Fernandez[15], Carlos Lopez-Larrea[3,6,8,12], Eliecer Coto[3,6,10,12],
Guillermo M Albaiceta[1,2,3,4,12]*

[1]Unidad de Cuidados Intensivos Cardiológicos. Hospital Universitario Central de Asturias, Oviedo, Spain; [2]Centro de Investigación Biomédica en Red (CIBER)-Enfermedades Respiratorias. Instituto de Salud Carlos III, Madrid, Spain; [3]Instituto de Investigación Sanitaria del Principado de Asturias, Oviedo, Spain; [4]Instituto Universitario de Oncología del Principado de Asturias, Oviedo, Spain; [5]Unidad de Cuidados Intensivos Polivalente. Hospital Universitario Central de Asturias, Oviedo, Spain; [6]Red de Investigación Renal (REDINREN), Madrid, Spain; [7]Servicio de Inmunología. Hospital Universitario Central de Asturias, Oviedo, Spain; [8]Area de Gestión Clínica de Pediatría. Hospital Universitario Central de Asturias, Oviedo, Spain; [9]Department of Anesthesiology and Operative Intensive Care Medicine CCM/CVK, Charité - Universitätsmedizin Berlin, corporate member of Freie Universität Berlin, Humboldt Universität zu Berlin, and Berlin Institute of Health, Berlin, Germany; [10]Servicio de Genética Molecular. Hospital Universitario Central de Asturias, Oviedo, Spain; [11]Servicio de Microbiología. Hospital Universitario Central de Asturias, Oviedo, Spain; [12]Universidad de Oviedo, Oviedo, Spain; [13]Servicio de Informática. Hospital Universitario Central de Asturias, Oviedo, Spain; [14]Critical Care Center, Hospital Universitari Parc Taulí, Institut D'Investigació I Innovació Parc Taulí I3PT, Universitat Autònoma de Barcelona, Sabadell, Spain; [15]Servicio de Bioquímica Clínica. Hospital Universitario Central de Asturias, Oviedo, Spain

*For correspondence:
gma@crit-lab.org

*These authors contributed equally to this manuscript.

Competing interest: The authors declare that no competing interests exist.

## Abstract

**Background:** Variants in *IFIH1*, a gene coding the cytoplasmatic RNA sensor MDA5, regulate the response to viral infections. We hypothesized that *IFIH1* rs199076 variants would modulate host response and outcome after severe COVID-19.

**Methods:** Patients admitted to an intensive care unit (ICU) with confirmed COVID-19 were prospectively studied and rs1990760 variants determined. Peripheral blood gene expression, cell populations, and immune mediators were measured. Peripheral blood mononuclear cells from healthy

volunteers were exposed to an MDA5 agonist and dexamethasone ex-vivo, and changes in gene expression assessed. ICU discharge and hospital death were modeled using rs1990760 variants and dexamethasone as factors in this cohort and in-silico clinical trials.

**Results:** About 227 patients were studied. Patients with the *IFIH1* rs1990760 TT variant showed a lower expression of inflammation-related pathways, an anti-inflammatory cell profile, and lower concentrations of pro-inflammatory mediators. Cells with TT variant exposed to an MDA5 agonist showed an increase in *IL6* expression after dexamethasone treatment. All patients with the TT variant not treated with steroids survived their ICU stay (hazard ratio [HR]: 2.49, 95% confidence interval [CI]: 1.29–4.79). Patients with a TT variant treated with dexamethasone showed an increased hospital mortality (HR: 2.19, 95% CI: 1.01–4.87) and serum IL-6. In-silico clinical trials supported these findings.

**Conclusions:** COVID-19 patients with the *IFIH1* rs1990760 TT variant show an attenuated inflammatory response and better outcomes. Dexamethasone may reverse this anti-inflammatory phenotype.

**Funding:** Centro de Investigación Biomédica en Red (CB17/06/00021), Instituto de Salud Carlos III (PI19/00184 and PI20/01360), and Fundació La Marató de TV3 (413/C/2021).

## Editor's evaluation

Authors present an interesting approach of COVID-19 multiorgan failure and loss of homeostasis attributed to modified adaptive immune response due to presence/absence of MDA5 polymorphism(rs1990760). Data presented by authors are very interesting representing novel scientific work that adds new and potentially pivotal information to existing knowledge. Key data presented strongly support an individualized approach during COVID-19 pandemic that could eventually alter therapeutic algorithm regarding dexamethasone administration implemented for these patients.

## Introduction

The spectrum of disease after infection by SARS-CoV-2 (COVID-19) may range from mild respiratory symptoms to a severe form of lung injury fulfilling acute respiratory distress syndrome (ARDS) criteria (*Grasselli et al., 2020*). In these severe cases, systemic response to infection may be associated with multiorgan failure and death (*Du et al., 2020*). Therefore, outcomes in critically ill COVID-19 patients are related not only to viral clearance, but also to preservation of homeostasis.

The mechanisms responsible for the development of severe forms of COVID-19 have not been fully elucidated, but non-adaptive inflammatory responses play a central role. The only strategies that have decreased mortality in this population, steroids (*WHO Rapid Evidence Appraisal for COVID-19 Therapies (REACT) Working Group et al., 2020*) and blockade of the IL-6 pathway (*REMAP-CAP Investigators et al., 2021*), aim to limit this exacerbated immune response to prevent organ dysfunction.

The human gene *IFIH1*, located in the reverse strand of chromosome 2, encodes MDA5, a helicase that acts as a cytoplasmatic virus receptor. After binding to a viral RNA strand, MDA5 interacts with a mitochondrial adapter (MAVS, mitochondrial antiviral signaling protein), triggering the transcription of type-1 interferon genes and ultimately the systemic inflammatory response. In humans, the rs1990760 polymorphism encodes a variant of the *IFIH1* gene (NM_022168: c.2836G > A [p.Ala946Thr]) that has been related to different susceptibility to viral infections and autoimmune disorders (*Gorman et al., 2017*). By regulation of IFN-dependent pathways, *IFIH1* participates in a feedback loop that ultimately modulates viral clearance and host inflammatory responses. In experimental models of Coxsackie virus infection, TT variants in rs1990760 result in lower pro-inflammatory cytokine levels without a major reduction in viral clearance (*Domsgen et al., 2016*).

Although the role of this variant in Coronavirus infections has not been explored, it has been proposed that the rs1990760 TT variant could confer resistance to SARS-CoV-2 infection and that differences in allelic frequencies could explain the epidemiological features of the pandemic in different countries (*Maiti, 2020*). We hypothesized that, once infection is established, the inflammatory response in severe COVID-19 patients could be conditioned by *IFIH1* variants. To test this hypothesis, we prospectively followed a cohort of critically ill patients with confirmed infection by

**eLife digest** Patients with severe COVID-19 often need mechanical ventilation to help them breathe and other types of intensive care. The outcome for many of these patients depends on how their immune system reacts to the infection. If the inflammatory response triggered by the immune system is too strong, this can cause further harm to the patient.

One gene that plays an important role in inflammation is *IFIH1* which encodes a protein that helps the body to recognize viruses. There are multiple versions of this gene which each produce a slightly different protein. It is possible that this variation impacts how the immune system responds to the virus that causes COVID-19.

To investigate, Amado-Rodríguez, Salgado del Riego et al. analyzed the *IFIH1* gene in 227 patients admitted to an intensive care unit in Spain for severe COVID-19 between March and December 2020. They found that patients with a specific version of the gene called TT experienced less inflammation and were more likely to survive the infection.

Physicians typically treat patients with moderate to severe COVID-19 with corticosteroid drugs that reduce the inflammatory response. However, Amado-Rodríguez, Salgado del Riego et al. found that patients with the TT version of the *IFIH1* gene were at greater risk of dying if they received corticosteroids.

The team then applied the distribution of *IFIH1* variants among different ethnic ancestries to data from a previous clinical trial, and simulated the effects of corticosteroid treatment. This 'mock' clinical trial supported their findings from the patient-derived data, which were also validated by laboratory experiments on immune cells from individuals with the TT gene.

The work by Amado-Rodríguez, Salgado del Riego et al. suggests that while corticosteroids benefit some patients, they may cause harm to others. However, a real-world clinical trial is needed to determine whether patients with the TT version of the *IFIH1* gene would do better without steroids.

SARS-CoV-2, in which peripheral blood gene expression, cell populations, concentrations of immune mediators, and clinical outcomes were studied and related to rs1990760 variants.

## Materials and methods

### Study design

This single-center prospective, observational study was approved (ref. 2020/188) by the Clinical Research Ethics Committee of Principado de Asturias (Spain). Informed consent was obtained from each participant or next of kin. Given the exploratory nature of the study objective and the absence of previous data, no formal sample size calculations were performed. The study started after approval from the ethics committee and finished in December 2020, after the second pandemic wave.

All patients with confirmed SARS-CoV-2 infection (*Escudero et al., 2020*) and meeting the Kigali modification (*Riviello et al., 2016*) of ARDS criteria (to include patients without mechanical ventilation) from March 16, 2020 to December 10, 2020 were included in the study. Exclusion criteria were age <18, any condition that could explain the respiratory failure other than COVID-19, do not resuscitate orders or terminal status, or refusal to participate. Patients were followed until hospital discharge, and clinical and analytical data collected. The main outcome was intensive care unit (ICU) discharge alive and spontaneously breathing. Secondary outcome was hospital discharge.

All biochemical analyses (cytokine measurements, RNAseq, estimation of cell populations…, see below) were performed by researchers blinded to genotype and outcome.

### SARS-CoV-2 detection and quantification

The presence of SARS-CoV-2 was analyzed by detecting viral genome using a multiple quantitative retrotranscriptase (RT)-PCR. Nucleic acids were purified by MagNa Pure 96 System (Roche, Geneva, Switzerland) from the swabs transport medium. The extracts were subjected to an amplification reaction using TaqMan Fast 1-Step Master Mix (Life Technologies, Carlsbad, CA) supplemented with a mixture of primers (Thermo Fisher Scientific, Walthan, MA) and TaqMan MGB probes (Applied Biosystems, Foster City, CA) directed against ORF1ab and N genes (*Supplementary file 1a*). Amplifications

and subsequent analysis were carried out using the Applied Biosystems 7500 Real-Time PCR System (*Escudero et al., 2020*). Amplification of viral genes with a Ct number lower than 35 was considered positive. Viral load was normalized by the number of cells and expressed as copies/1000 cells, as previously described (*Gómez-Novo et al., 2018*). Clearance of SARS-CoV-2 was evaluated by fitting viral load in tracheobronchial samples after its peak value over time using an exponential decay function and calculation of viral clearance half-life ($\lambda_{\text{Viral Clearance}}$).

### Genotyping

DNA was extracted from total blood leukocytes (1 ml) with an automated equipment (Promega Maxwell). *IFIH1* rs1990760 C/T polymorphism determined by real-time PCR with TaqMan genotyping master mix (Life Technologies) and TaqMan probes (Thermo Fisher Scientific, assay C_2780299_30) in an ABI-7500 device. The genotyping strategy was validated by Sanger sequencing of selected individuals from the three genotypes. Based on previous data showing that patients with a TT genotype have a different immune response compared to CC and CT (*Wawrusiewicz-Kurylonek et al., 2020*; *Zhang et al., 2018*), patients with these two variants were grouped and compared against the TT group.

### Blood sampling

Two blood samples were taken in the first 24 hr after ICU admission. No patient had received steroids at the time of sampling. About 3 ml of blood was collected in Tempus tubes (Thermo Fisher Scientific) for RNA isolation and immediately stored at –80°C. Additional 5 ml was collected in a Vacutainer serum tube (BD Biosciences), and isolated by centrifugatio, and stored at –80°C until analysis.

### RNAseq

After thawing, blood from Tempus tubes was diluted in phosphate-buffered saline (PBS) (1:3 v/v) and centrifuged at 3000 rpm for 30 min. Supernatant was discarded and the pellet resuspended in TRIzol (Sigma-Aldrich, Poole, UK) and precipitated overnight with isopropanol at –20°C. After centrifugation, RNA pellets were washed with 70% ethanol and resuspended in RNAse-free water. RNA quality was assessed using a TapeStation, and only samples with a RIN (RNA integrity number) above 8 were analyzed.

RNA sequencing was performed using Ion AmpliSeq Transcriptome Human Gene Expression Kit, in an Ion S5 GeneStudio sequencer (Ion Torrent). Briefly, 10 ng of total RNA was retrotranscribed and the obtained cDNA used for library synthesis using Ion AmpliSeq Transcriptome kits to amplify all the canonical human transcripts. After template preparation in an automated Ion Chef Instrument, semiconductor 540 chips were run in an Ion S5 GeneStudio sequencer. Torrent Suite software was used for base calling, alignment, and sequence quality controls. The generated FASTQ files (available at Gene Expression Omnibus, accession numbers GSE168400 and GSE 177025) were mapped against a reference transcriptome (obtained from http://refgenomes.databio.org) and transcripts counted using Salmon v1.4 software (*Patro et al., 2017*). Raw counts were compared between genotypes using the DESeq2 library (*Love et al., 2014*). The $\log_2$-fold change between variants for each gene and the adjusted p-value (corrected using a false discovery rate of 0.05) were calculated and analyzed using Ingenuity Pathway Analysis (Qiagen, USA) to identify overrepresented gene sets and networks. Over the identified network, in-silico effects of *IFIH1* downregulation and addition of exogenous dexamethasone were performed.

### Peripheral blood cell populations

Circulating cell populations were estimated from gene expression using a previously validated deconvolution algorithm (*Vallania et al., 2018*). Using a reference expression matrix, proportions of 20 different cell lines were calculated. Only cell lines present in more than five samples were considered. As a quality check, we analyzed the correlation between estimated and measured lymphocyte percentages. The obtained correlation coefficient was 0.61 (*Figure 3—figure supplement 1*). It must be noted, however, that the deconvolution method estimates proportions over transcriptionally active cells, which may not be equivalent to the obtained cell count (as there may be inactive cells in the latter).

### Inflammatory mediators

A panel of inflammatory mediators was studied in serum from patients not receiving steroids during their first ICU day. Serum concentrations of interferons (IFN)-β, -γ and -$\lambda$, tumor necrosis factor

(TNF)-α, interleukins (IL)-1β and -6, and chemokines CXCL8, CXCL9, CXCL10, CXCL16, CCL2, CCL3, CCL4, and CCL7 were measured using a multiplexed assay (Luminex custom panel). Concentrations below the lower limit of detection for a given mediator were replaced with half of that limit.

## Ex-vivo experiments

To study the potential interferences between *IFIH1* rs1990760 variants and dexamethasone predicted by in-silico analyses, an ex-vivo experiment was designed. Blood samples from healthy volunteers (genotyped for *IFIH1* rs1990760 variants using DNA obtained from buccal swabs) were collected in EDTA tubes and immediately processed. Peripheral blood mononuclear cells (PBMCs) were isolated via density-gradient centrifugation with Lymphoprep (Axis-Shield PoC AS, Oslo, Norway). Cells were washed with red blood cell lysis buffer (NH$_4$Cl 0.1 M, KHCO$_3$ 0.01 M, EDTA 0.1 mM in dH$_2$O, pH 7.33) and PBS before being resuspended in RPMI-1640 (Gibco, USA) + 10% fetal bovine serum (FBS). Cells were seeded in 12-well plates at a final concentration of 1.5×10$^6$ PBMC/ml and cultured at 37°C and 5% CO$_2$ in presence of medium/FBS, medium/FBS plus a MDA5 ligand (1 µg/ml high-molecular weight poly-I:C/LyoVec, Invivogen, USA), or medium plus the MDA5 ligand and 10 µM dexamethasone (Kern Pharma, Spain). Although poly-I:C may bind to both RIG-I and MDA-5, long (high-molecular weight) poly-I:C binds specifically to MDA-5 (*Kato et al., 2008*). After 24 hr, cells were collected and homogenized in TRIzol for RNA extraction. 500 ng of total RNA was retrotranscribed into cDNA using an RT-PCR Kit (High-capacity cDNA rt Kit, Applied Biosystems, USA). Expression of *STAT1, STAT3, FOXO3, IL6,* and *GAPDH* was quantified using 5 ng of cDNA per well and in triplicate for each sample. SYBR Green Power up (Thermo Fisher Scientific) and 10 µM of the corresponding primers (*Supplementary file 1b*) were used in all the experiments. The relative expression of each gene was calculated as $2^{-\Delta CT(\text{gene of interest}) - \Delta CT(GAPDH)}$.

## Clinical trial simulations

Data from the RECOVERY trial were extracted and used to estimate risk ratios (RRs) for each rs1990760 variant, with and without steroids (see online Appendix for details). With these data, a survival model was developed and used to simulate scenarios with different allelic frequencies and baseline risks of death. Hazard ratios (HRs) for 28-day mortality were calculated from these simulations.

## Statistical analysis

Data are shown as median (interquartile range). All data points and sample sizes represent individual patients/biological replicates. Comparisons between *IFIH1* variants were done using Wilcoxon or ANOVA tests, and p-values corrected using a false discovery rate of 0.05. Results from the ex-vivo model were fitted to a mixed-effects model including experimental group and genotype as covariables, and post hoc comparisons evaluated using Holm's correction. No outliers were discarded.

Survival was analyzed using competing risks, Cox regression model, with ICU/hospital discharge alive and spontaneously breathing and death as competing risks, using the Aalen-Johansen estimator, as previously described (*Barbaro et al., 2020*). This competing-risks framework is needed as patients discharged alive have a low probability of death, so censoring at the time of discharge using a standard Kaplan-Meier approach would lead to biased observation, as the probability of death is different than in those still followed (i.e., informative censoring). Patients with an rs1990760 CC/CT variant not treated with steroids were considered the reference category in all the analyses.

All the analyses and plots were performed using the R 4.0.1 statistical environment (*R Development Core Team, 2020*) with the packages data.table (*Dowle and Srinivasan, 2021*), multcomp (*Hothorn et al., 2008*), survival (*Therneau, 2020*), MetaIntegrator (*Haynes et al., 2017*), and ggplot2 (*Wickham, 2016*, p. 2).

## Results

### Study cohort

About 250 patients were admitted in ICU due to suspected or confirmed COVID-19. Among these, 227 were included in the study. The study flow chart and reasons for exclusion are shown in *Figure 1*. Basic demographic and clinical data are shown in *Table 1*. *IFIH1* rs1990760 variants in this population met Hardy-Weinberg conditions (53 CC, 110 CT, 64 TT, chi-square p=0.19). There were no differences

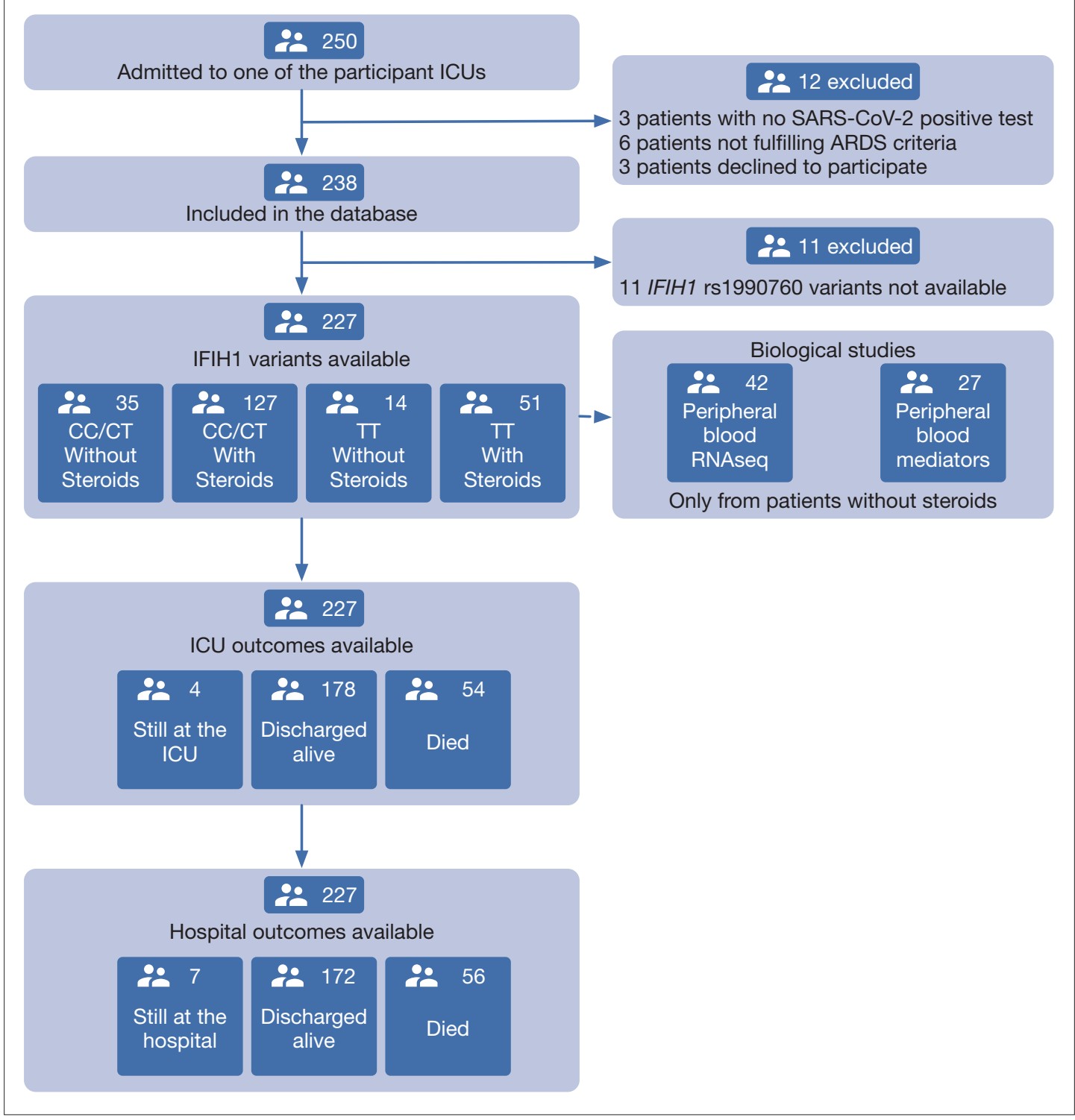

**Figure 1.** Patient study flow chart. From 250 patients admitted to the participant ICUs during the study period, 227 were included in the study and classified according to the *IFIH1* rs1990760 variants and prescription of steroid therapy. Peripheral blood samples for gene expression (N=41) and quantification of immune mediators (N=28) were taken in the first day after ICU admission in patients not receiving steroids. All patients were followed up until death or hospital discharge.

**Table 1.** Clinical characteristics of the study cohort.

COPD: Chronic Obstructive Pulmonary Disease. Values are shown as absolute count or median (interquartile range). PBW: Predicted body weight. NIV: Non-invasive ventilation. PEEP: Positive End-Expiratory Pressure. *p-values calculated for proportion over the number of intubated patients.

| | rs1990760 | | | |
| --- | --- | --- | --- | --- |
| | Overall (n=227) | TT (n=65) | CC/CT (n=162) | p-value |
| Demographics | | | | |
| Age (years) | 67 (60–75) | 68.5 (63–75.25) | 66 (58–75) | 0.123 |
| Sex | | | | |
| Male | 174 (77) | 47 (72) | 127 (78) | 0.327 |
| Female | 53 (23) | 18 (28) | 35 (22) | |
| Race | | | | |
| Black | 3 (1.5) | 1 (2) | 2 (1.5) | |
| White | 207 (91) | 62 (95) | 146 (90) | |
| Latino | 15 (7) | 2 (3) | 13 (8) | |
| Asian | 1 (0.5) | 0 | 1 (0.5) | 0.527 |
| Body mass index (kg/m$^2$) | 30 (27–33) | 30 (26–33) | 30 (27–32) | 0.592 |
| Days since symptom onset | 8 (6–11) | 8 (6–10) | 8 (6–11) | 0.873 |
| Days from hospital admission to ICU admission | 2 (1–4) | 2 (1–3) | 2 (1–4) | 0.841 |
| APACHE-II score | 15 (12–18) | 15.5 (13–18) | 14 (11—18) | 0.177 |
| Comorbidities | | | | |
| Arterial hypertension | 132 (58) | 38 (59) | 94 (58) | 0.932 |
| Diabetes | 51 (23) | 14 (22) | 37 (23) | 1 |
| Chronic kidney disease | 15 (7) | 5 (8) | 10 (6) | 0.743 |
| COPD | 17 (8) | 7 (11) | 10 (6) | 0.339 |
| Cirrhosis | 2 (1) | 1 (2) | 1 (0.5) | 1 |
| Neoplasms | | | | |
| No | 215 (95) | 62 (95)3 (5)0 | 154 (95) | |
| Active | 10 (4) | 3 (5) | 7 (3.5) | |
| Past | 2 (1) | 0 | 2 (1.5) | 0.669 |
| Immunosuppresive drugs (incl. steroids) | 5 (2) | 3 (5) | 2 (1.5) | 0.142 |
| Lung function at ICU admission | | | | |
| Ventilation at admission | | | | |
| Spontaneous /NIV | 38 (17) | 11 (17) | 27 (17) | |
| Controlled invasive ventilation | 187 (82.5) | 53 (82) | 135 (83) | |
| Pressure support ventilation | 1 (0.5) | 1 (2) | 0 | 0.278 |
| FiO$_2$ | 0.5 (0.4–0.6) | 0.5 (0.4–0.7) | 0.5 (0.4–0.6) | 0.104 |
| PaO$_2$/FiO$_2$ (mmHg) | 204 (155–267) | 185 (142–226) | 218 (162–282) | 0.007 |
| PaCO$_2$ (mmHg) | 43 (39–47) | 44 (39–47) | 43 (39–48) | 0.924 |
| Respiratory rate (min$^{-1}$) | 18 (16–20) | 18 (16–20) | 18 (16–22) | 0.929 |

*Table 1 continued on next page*

*Table 1 continued*

| | rs1990760 | | | |
| --- | --- | --- | --- | --- |
| | Overall (n=227) | TT (n=65) | CC/CT (n=162) | p-value |
| Arterial pH | 7.38 (7.32–7.41) | 7.38 (7.33–7.42) | 7.37 (7.32–7.41) | 0.805 |
| Tidal volume/PBW (ml/Kg) | 7.6 (6.9–8.4) | 8 (6.9–9.2) | 7.5 (7–8.1) | 0.109 |
| Plateau pressure (cmH$_2$O) | 24 (21–28) | 24 (22–29) | 24 (21–27) | 0.221 |
| PEEP (cmH$_2$O) | 12 (10–14) | 12 (10–14) | 12 (10–14) | 0.397 |
| Driving pressure (cmH$_2$O) | 12 (10–14) | 12 (10–14) | 12 (10–14) | 0.496 |
| Respiratory system compliance (ml/cmH$_2$O) | 38 (32–50) | 36 (31–44) | 42 (32–50) | 0.11 |
| Laboratory results | | | | |
| Leukocytes (×10$^3$/µl) | 8.34 (6.05–11.38) | 8.77 (5.96–11.52) | 8.29 (6.08–11.18) | 0.773 |
| Lymphocytes (×10$^3$/µl) | 0.65 (0.48–0.93) | 0.62 (0.47–0.94) | 0.67 (0.49–0.93) | 0.266 |
| Serum creatinine (mg/dl) | 0.83 (0.62–1.13) | 0.87 (0.62–1.14) | 0.82 (0.62–1.11) | 0.859 |
| Serum ferritin (ng/ml) | 1107 (717–1667) | 965 (634–1263) | 1262 (776–2374) | 0.011 |
| D-dimer (ng/ml) | 1053 (681–1967) | 1085 (643–1805) | 1036 (683–2126) | 0.854 |
| Additional treatments at ICU admission | | | | |
| Steroids | 178 (78) | 51 (79) | 127 (78) | 1 |
| Vasoactive drugs | 102 (45) | 27 (42) | 75 (46) | 0.686 |
| Invasive mechanical ventilation | 210 (93) | 59 (91) | 151 (93) | 0.724 |
| Neuromuscular blocking agents | 99 (44) | 30 (46) | 69 (43) | 0.733* |
| Prone ventilation | 119 (52) | 32 (49) | 87 (54) | 0.657* |
| ECMO | 5 (2) | 2 (3) | 3 (2) | 0.544* |

in comorbidities or clinical data at admission between genotypes other than a lower PaO$_2$/FiO$_2$ ratio in patients with a TT variant (**Table 1**).

## IFIH1 rs1190760 variants and inflammatory response to SARS-CoV-2

Gene expression in peripheral blood during the first day of ICU admission was profiled in 42 patients who did not receive steroid therapy at that time (11, 19, and 12 with rs1990760 TT, CT, and CC genotypes, respectively). Expression of *IFIH1* was significantly lower in patients with the TT genotype (**Figure 2A** and **Figure 2—figure supplement 1**). Comparison of peripheral blood gene expression between patients with TT and CT/CC variants yielded significant differences in 179 genes (**Figure 2B–C**, **Supplementary file 2**). Visual inspection of the heatmap reveals that differences are quantitative rather than qualitative. Ingenuity pathway analysis revealed several gene networks involved in the regulation of the inflammatory response among the differentially expressed genes (**Figure 2D** and **Figure 2—figure supplement 2**). In-silico predictions suggest that *IFIH1* downregulation (such as in patients with the rs1990760 TT variant) decreases the expression of pro-inflammatory pathways (**Figure 2—figure supplement 3**).

Then we assessed the immunological consequences of these differences in gene expression and cell populations. There were no statistically significant differences in inferred percentages of classic CD14+ monocytes, circulating plasma cells, M2 macrophages, and CD56dim NK cells, between genotypes. Patients with the TT genotype showed an increase in hematopoietic precursors and myeloid dendritic cells (**Figure 3**).

Serum cytokines were measured at ICU admission in 28 patients (8, 10, and 10 with TT, CT, and CC genotypes, respectively). There were no differences in any of the measured interferons, which were below the limit of detection in a large proportion of patients. Patients with the TT genotype showed lower levels of pro-inflammatory mediators, including IL-6, CXCL10, CXCL16, and CCL7 (**Figure 4**).

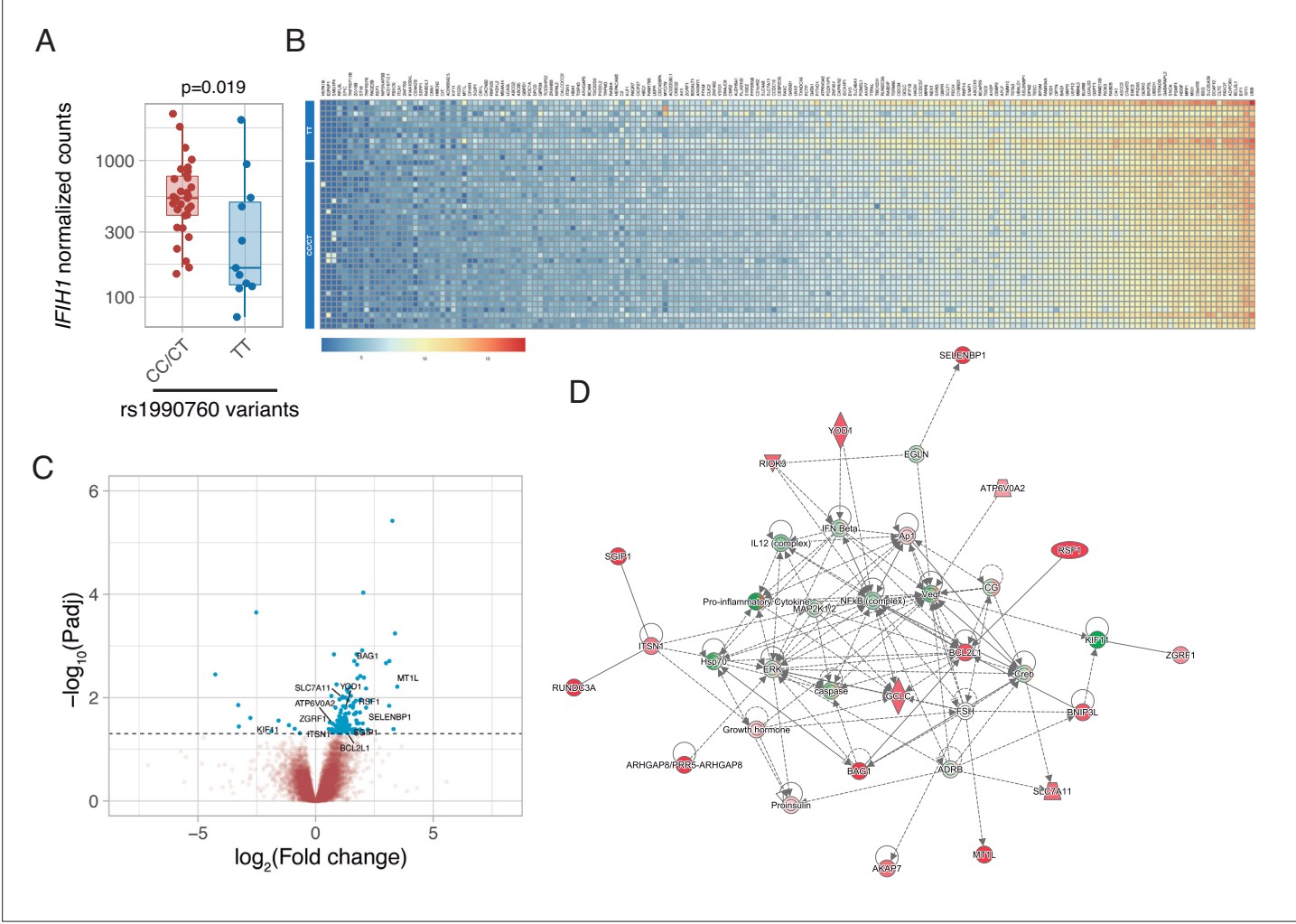

**Figure 2.** Differences in gene expression according to *IFIH1* rs1990760 genotypes. (**A**) Expression of IFIH1 in peripheral blood in patients with CC/CT (n=31) and TT (n=11) variants (p-value calculated using a Wilcoxon test). (**B**) Heatmap showing expression of the 160 genes with significant differences between variants. (**C**) Volcano plot showing the distribution of the magnitude of the differences in gene expression (Log₂ Fold change) and their statistical significance. Inflammation-related genes with differential expression and included in the network shown in panel (**D**) are labeled. (**D**) Inflammation-related gene network identified using Ingenuity Pathway Analysis on the RNAseq data. Points represent individual patient data. In boxplots, bold line represents the median, lower and upper hinges correspond to the first and third quartiles (25th and 75th percentiles) and upper and lower whiskers extend from the hinge to the largest or smallest value no further than 1.5 times the interquartile range.

The online version of this article includes the following figure supplement(s) for figure 2:

**Figure supplement 1.** *IFIH1* expression per rs1990760 variants.

**Figure supplement 2.** Gene networks involved in the regulation of the inflammatory response identified by Ingenuity Pathway Analysis among the genes with differential expression in patients with rs1990760 CC/CT and TT variants.

**Figure supplement 3.** In-silico predictions of IFIH1 downregulation over a gene network involved in the regulation of the inflammatory response.

There were no significant differences in any of the measured cytokines between patients with CC and CT variants (*Figure 4—figure supplement 1*).

## Ex-vivo effects of dexamethasone in IFIH1 variants

During the study period, dexamethasone was added to COVID-19 treatment based on results from published trials (***WHO Rapid Evidence Appraisal for COVID-19 Therapies (REACT) Working Group et al., 2020***). An in-silico analysis focused on the interactions between *IFIH1* and steroids suggested that dexamethasone could disrupt some of the effects caused by *IFIH1* downregulation

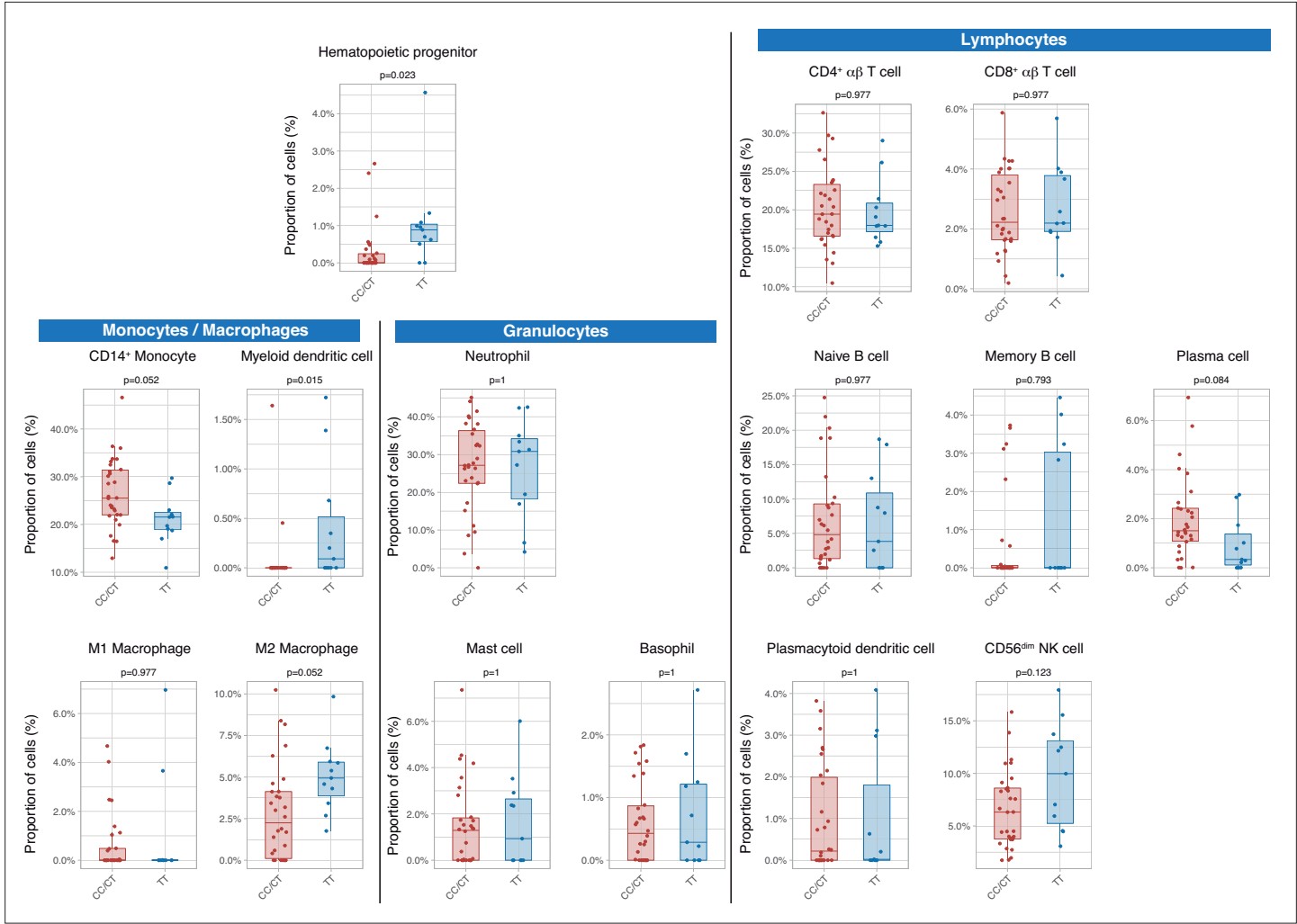

**Figure 3.** Differences in circulating cell populations according to *IFIH1* rs1990760 variants (31 and 11 samples from patients with a CC/CT or TT variant, respectively). Proportions of each cell line were estimated by deconvolution of RNAseq data. P-values were calculated using a Wilcoxon test and adjusted using the Benjamini-Hochberg method for a false discovery rate of 5%. Points represent individual patient data. In boxplots, bold line represents the median, lower and upper hinges correspond to the first and third quartiles (25th and 75th percentiles) and upper and lower whiskers extend from the hinge to the largest or smallest value no further than 1.5 times the interquartile range.

The online version of this article includes the following source data and figure supplement(s) for figure 3:

**Source data 1.** Raw data used in *Figure 3*, showing different peripheral cell populations obtained after deconvolution of RNAseq data.

**Figure supplement 1.** Correlation between measured and estimated (from deconvolution analysis) lymphocyte percentages in peripheral blood.

(*Figure 5—figure supplement 1*). Specifically, dexamethasone may change expression of several *IFIH1*-dependent genes, including *STAT1*, *STAT3*, or *FOXO3* among others.

To test these predictions, an ex-vivo experiment using PBMCs from healthy volunteers with different *IFIH1* rs1990760 variants (n=5, 6, and 7 for CC, CT, and TT variants, respectively) was performed. Cells were collected and exposed to poly I:C, to mimic SARS-Cov-2 infection, and dexamethasone. Compared to exposure to poly I:C alone, dexamethasone had no effect on *STAT1* (*Figure 5A*) or *STAT3* (*Figure 5B*) expression. However, the steroid increased the expression of *FOXO3* (*Figure 5C*) and *IL6* (*Figure 5D*) only in cells with the TT variant. These results suggest that dexamethasone may alter the inflammatory response triggered by MDA5 activation in those patients with the TT variant of the gene.

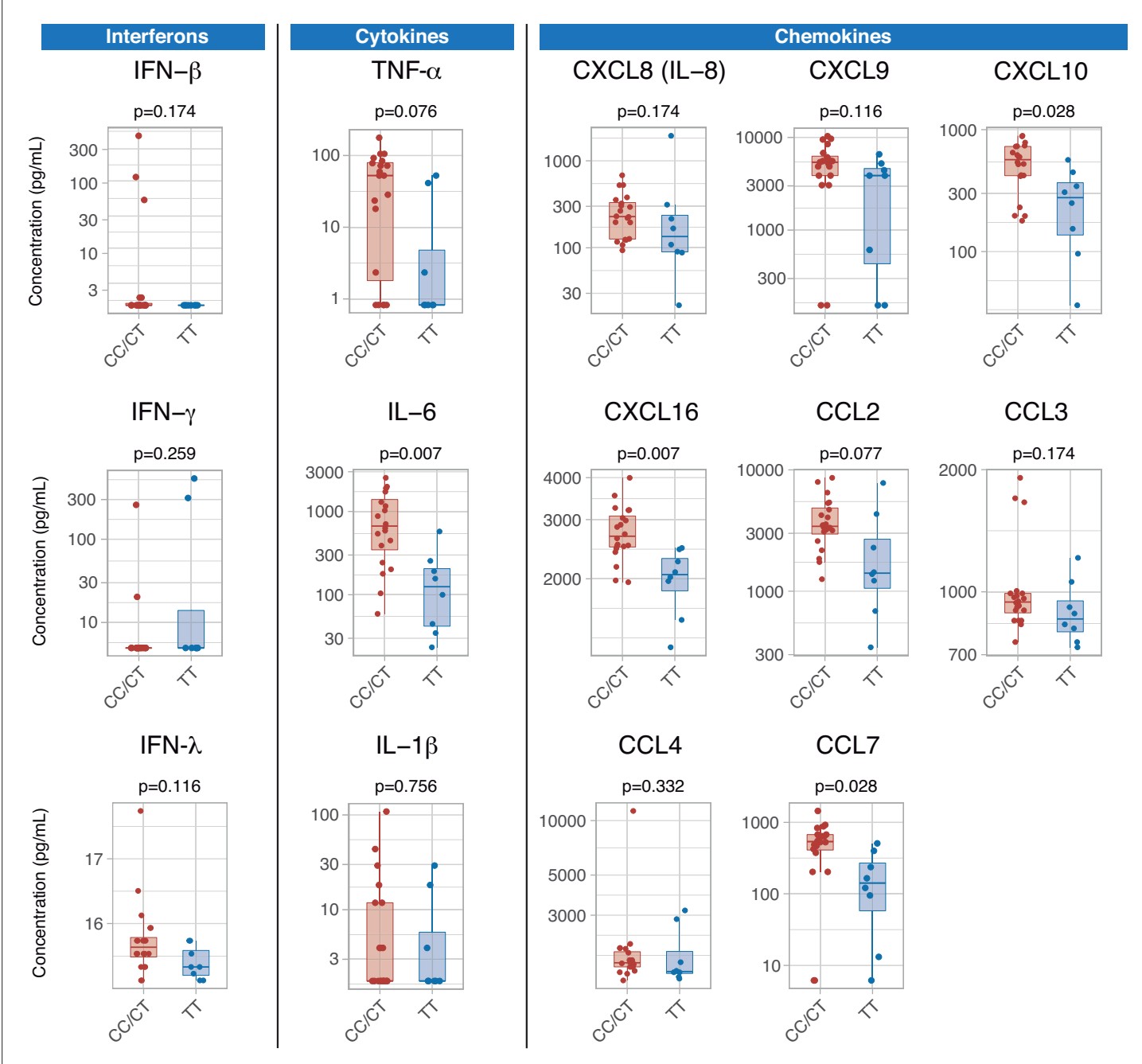

**Figure 4.** Differences in serum immune mediators according to *IFIH1* rs1990760 variants (19 and 8 samples from patients with a CC/CT or TT variant, respectively). P-values were calculated using a Wilcoxon test and adjusted using the Benjamini-Hochberg method for a false discovery rate of 5%. Points represent individual patient data. In boxplots, bold line represents the median, lower and upper hinges correspond to the first and third quartiles (25th and 75th percentiles) and upper and lower whiskers extend from the hinge to the largest or smallest value no further than 1.5 times the interquartile range.

The online version of this article includes the following source data and figure supplement(s) for figure 4:

**Source data 1.** Raw data used in *Figure 4*, showing serum concentration of inflammatory mediators.

**Figure supplement 1.** Serum inflammatory mediators in patients with a CC or CT variant in rs1990760 polymorphisms.

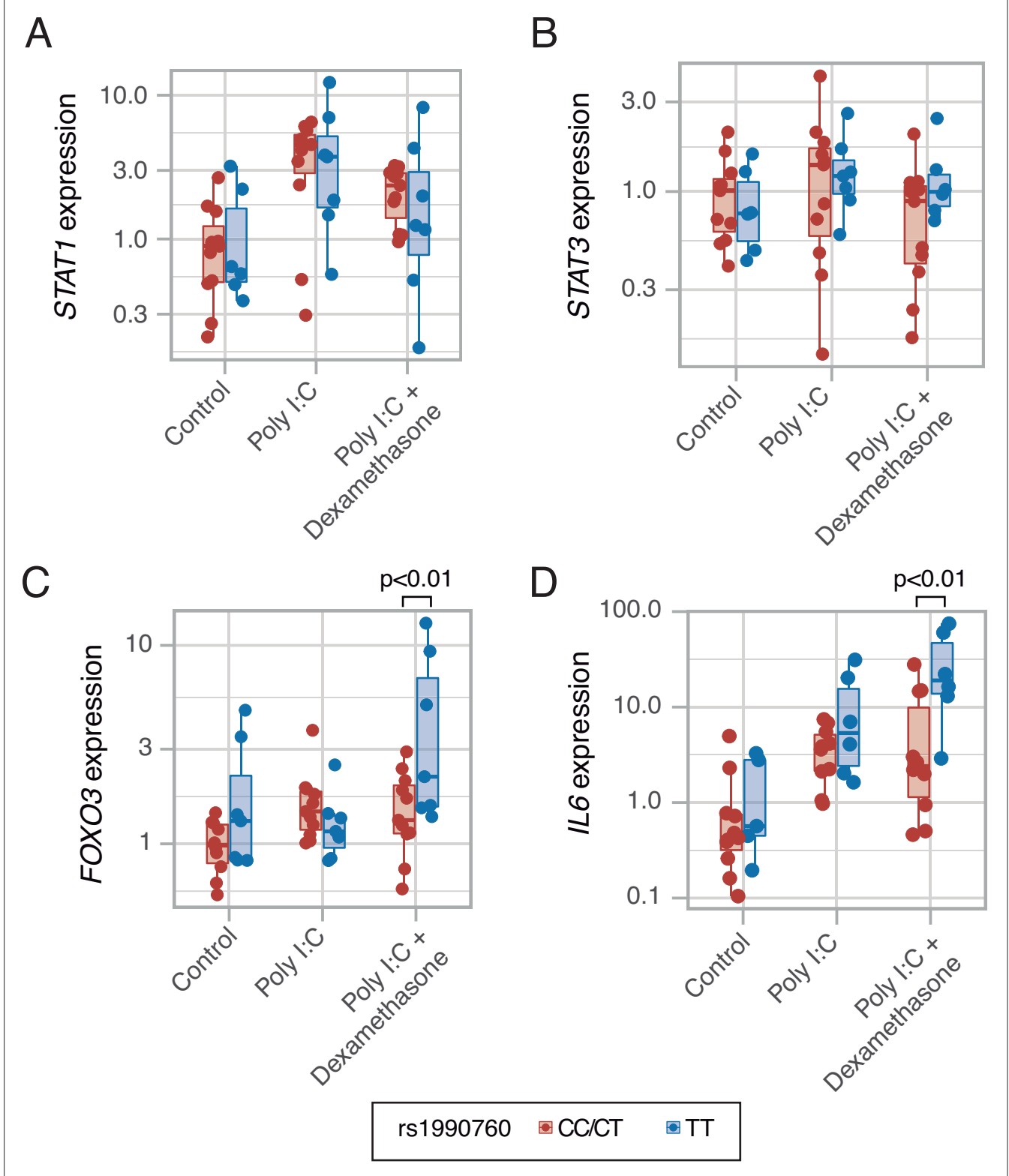

**Figure 5.** Expression of transcription factors *STAT1* (**A**, p=0.012, no significant differences between variants in post hoc tests), *STAT3* (**B**, p=0.443) and *FOXO3* (**C**, p=0.002), and cytokine *IL6* (**D**, p<0.001) induced by MDA5 activation using high molecular weight poly-I:C (a viral RNA analog) and dexamethasone in peripheral mononuclear blood cells. Results were fitted using a mixed-effects linear model including cell donor as a fixed effect and experimental group and genotype as random effects. Pairwise p-values lower than 0.05 (using Holm's correction) are shown. Points represent individual patient data. In boxplots, bold line represents the median, lower and upper hinges correspond to the first and third quartiles (the 25th and 75th

*Figure 5 continued on next page*

*Figure 5 continued*

percentiles) and upper and lower whiskers extend from the hinge to the largest or smallest value no further than 1.5 times the interquartile range.

The online version of this article includes the following source data and figure supplement(s) for figure 5:

**Source data 1.** Raw data used in *Figure 5*, showing gene expression in an ex-vivo model of MDA5 stimulation.

**Figure supplement 1.** Predicted effects of IFIH1 downregulation in absence (**A**) or in presence (**B**) of dexamethasone.

## Impact of IFIH1 variants on outcomes in critically ill COVID-19 patients

Median follow-up was 25 days (interquartile range 16–39 days) and 27 days in survivors (interquartile range 17–44 days). There were no differences in ICU length of stay among groups (*Supplementary file 1c*). ICU survival and hospital mortality were modeled using *IFIH* rs1990760 genotype and steroid treatment as interacting covariables. There were no significant differences in the main clinical characteristics among the resulting groups (*Supplementary file 1c*). Regarding ICU survival (*Figure 6A* and *Supplementary file 1c*), 27 out of 35 patients with CC/CT alleles who did not receive steroids were discharged alive and spontaneously breathing from the ICU (HR: 1, used as reference). In patients with this variant, dexamethasone was not related to a better outcome (98 out of 128 patients discharged, HR: 1.20 [0.78–1.38], p=0.41). All patients with the TT allele (n=14) who did not receive steroids were discharged alive (HR: 2.49 [1.29–4.79], p=0.012). Steroid treatment in patients with the TT variant was related to the loss of this benefit (32 out of 50 patients discharged alive, HR: 1.03 [0.62–1.72], p=0.91).

Hospital mortality is shown in *Figure 6B* and *Supplementary file 1c*. There were nine hospital deaths in patients with the CT/CC genotypes not treated with steroids (out of a total of 35 patients), and 27 out of 128 hospital deaths when treated with steroids, resulting in HR of 1.11 [0.52–2.37] (p=0.80). All patients with the TT allele who were not treated with steroids survived after their hospital stay (HR $1.23 \times 10^{-7}$, confidence intervals, and p-value cannot be computed due to the absence of events). Patients with the TT allele and treated with steroids showed a worse outcome (19 deaths in 50 patients, HR: 2.19 [1.01–4.87], p=0.05).

To investigate the causes responsible for the differences in mortality among groups, we first quantified viral load at diagnosis (*Figure 6C*) and clearance after its peak value (*Figure 6D*). There were no differences in these parameters related to *IFIH1* variants or treatment groups. We also compared serum IL-6 levels at ICU admission and 1 week later. Compared to patients with CC/CT variants not treated with steroids, patients with the same genotype but receiving dexamethasone and patients with the TT variant not receiving steroids showed lower levels of IL-6 at admission (*Figure 6E*). After 1 week (*Figure 6F*), serum IL-6 levels remained lower in patients receiving dexamethasone with the CC/CT variant and in those with the TT variant and not treated with this drug. However, IL-6 levels were higher in those with the TT variant and treated with steroids, in line with our ex-vivo findings.

To explore the mechanisms behind these differences, peripheral blood gene expression at ICU day 4 was compared between patients treated with or without steroids for each rs1990760 variant. *IFIH1* expression decreased in patients with rs1990760 CC/CT variants treated with steroids, but not in those with a TT variant (*Figure 6G*). In opposite, steroids increased *FOXO3* expression only in patients with a TT variant (*Figure 6H*), resembling the results from the ex-vivo experiments. *IL6* gene raw counts at day 4 were below 5 in all the patients, and thus not compared. When the whole transcriptomes were compared, steroids significantly changed the expression of 58 genes in patients with a CC/CT variant and 23 in patients with a TT variant (*Supplementary file 3*). Overall, changes in gene expression were qualitatively different between variants, with only three genes in common (*Figure 6I* and *Figure 6—figure supplement 1*).

## In-silico clinical trials

These findings may support the hypothesis that COVID-19 populations with a low proportion of the rs1990760 TT variant would show a better response to dexamethasone. To explore this prediction, data from the RECOVERY trial (*RECOVERY Collaborative Group et al., 2021*) were examined. The T-allele frequency in populations from a Black/Asian ancestry is 0.13 (data available at https://grasp.nhlbi.nih.gov/Covid19GWASResults.aspx), so only 1.7% has a TT variant. In these populations, RR was 0.7 (0.51–0.95), whereas in patients from a White ancestry (with a T allele frequency of 0.61), RR increased to 0.9 (0.8–1.2).

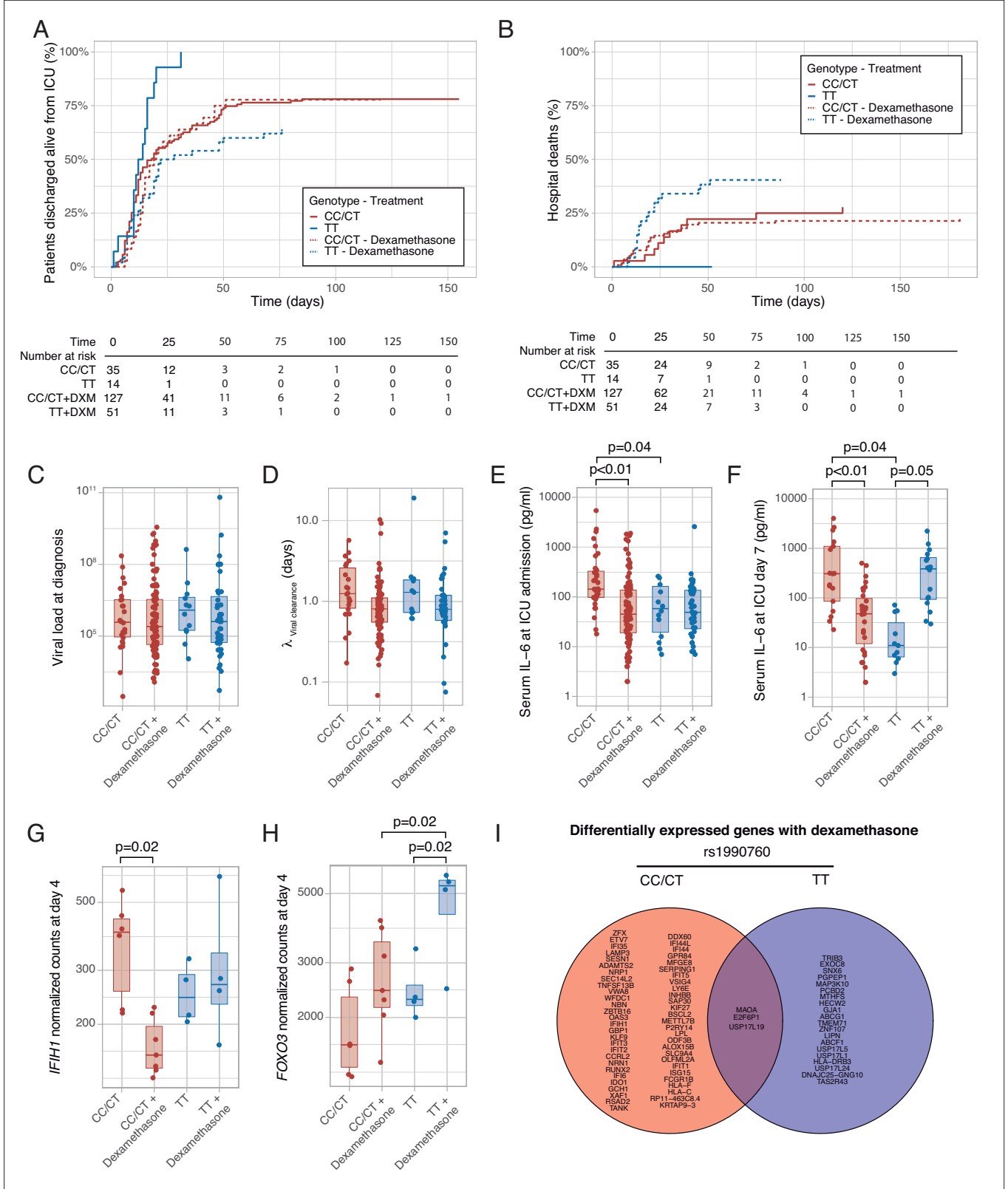

**Figure 6.** Outcomes according to *IFIH1* rs1990760 variants and treatment with dexamethasone. (**A**) Cumulative incidence of ICU discharge alive and spontaneously breathing for each group. (**B**) Cumulative incidence of hospital death for each group. Outcomes were modeled as competing events. (**C**) Viral load at diagnosis for each group (p=0.888). (**D**) Viral clearance evaluated as half-life of an exponential decay function calculated from the maximal viral load in each patient (p=0.056, no significant differences in post hoc tests). (**E, F**) Serum IL-6 concentrations during the first day (**E**, p<0.001) and

*Figure 6 continued on next page*

*Figure 6 continued*

7 days (**F**, p=0.014) after ICU admission for each group. Differences among groups in panels (**C–F**) were evaluated using an analysis of the variance. Pairwise p-values lower than 0.05 (using Holm's correction) are shown. (**G, H**) Changes in *IFIH1* (**G**) and *FOXO3* (**H**) expression in day 4 of ICU stay for each group (n=6 and 7 for CC/CT variants without and with dexamethasone, respectively, n = 4 and 4 for TT variant without and with dexamethasone, respectively). (**I**) Genes with dexamethasone-induced changes in gene expression in ICU day four in each rs1990760 variant. Points represent individual patient data. In boxplots, bold line represents the median, lower and upper hinges correspond to the first and third quartiles (25th and 75th percentiles) and upper and lower whiskers extend from the hinge to the largest or smallest value no further than 1.5 times the interquartile range.

The online version of this article includes the following figure supplement(s) for figure 6:

**Figure supplement 1.** Heatmaps with the differentially expressed genes in response to steroids in patients of each rs1990760 variant.

Using these data, RRs related to each rs1990760 variant, with and without steroids, were extracted and a survival model developed (see online supplement for details). A simulation including 500 patients from each rs1990760 variant and treatment arm (placebo or dexamethasone) showed a significant reduction in mortality with steroids in patients with a CC/CT variant, but not in those with a TT variant (*Figure 7A*). In patients not-receiving steroids, HRs related to a TT variant, compared to CC/CT variants, were independent of the allelic frequency (*Figure 7B*). Simulations of steroid therapy with different allelic frequencies showed that HRs in each specific variant remained constant whereas the effect of steroids in the overall population was dependent on the allele distribution (*Figure 7C* and *Figure 7—figure supplement 1*).

## Discussion

Our results show that critically ill patients with the TT variant in the *IFIH1* rs1990760 polymorphism have an attenuated inflammatory response to severe SARS-CoV-2 infection, leading to a decreased mortality. In this selected population, treatment with steroids has no immunomodulatory effects and could be related to worse outcomes. These results confirm the impact of the host response on patients' outcomes and suggest that patient geno/phenotypes should be taken into account to prescribe steroids in this setting.

Host response is a major determinant of outcome in critically ill patients, including those with COVID-19. Several genetic variants involved in the inflammatory response have been related to SARS-CoV-2 infection and its severity (*Bovijn et al., 2020*; *Gómez, 2021*; *Zhang et al., 2020*). Activation of immune responses causes local and systemic inflammation, aimed to block viral replication. However, exacerbation of these responses can cause organ damage even after virus clearance. Our results recapitulate previous findings in COVID-19 describing the release of pro-inflammatory mediators, NK cell exhaustion, monocyte dysregulation, and emergency hematopoiesis (*Schulte-Schrepping et al., 2020*; *Wen et al., 2020*; *Wilk et al., 2020*). Some of our findings in patients with the rs1990760 TT variant, including lower levels of circulating pro-inflammatory molecules and a shift toward anti-inflammatory cell populations (hematopoietic precursors, M2 macrophages, or CD56[dim] NK cells) have been linked to better outcomes in other observational studies (*Maucourant et al., 2020*).

Recognition of viral RNA by cytosolic receptors leading to the ultimate induction of expression of types I and II IFNs as part of the activation of innate antiviral signaling cascades, may not only relay on MDA5 receptor but also on other retinoid acid-inducible gene (RIG)-I-like receptors, or on the endosomal Toll-like receptor 3 (TLR3) (*Takeuchi and Akira, 2010*). Interestingly, no differences were found in serum interferons levels. This may be explained by several factors. Most patients included in our study had serum IFNs levels below the inferior detection threshold. This result is similar to the one reported by *Galani et al., 2021*, where IFN-lambda and type I IFN production were both diminished and delayed in moderate-to-severe COVID-19 patients followed up during hospitalization. In our study, IFNs levels were only measured on day 1 of ICU admission, precluding any conclusions regarding time-dependent changes on these mediators' levels during, for instance, subacute or long-term infection.

More importantly, transcriptomic analysis did not reveal significantly different expression levels for IFIH1 gene after correction for multiple comparisons, although single-gene comparison showed a downregulated expression in patients with TT genotype. Although expression levels may be similar, full activation of an antiviral response triggering specific signaling intermediates and transcription factors, may vary between genotypes due to differences in MDA5 molecular architecture and function. The

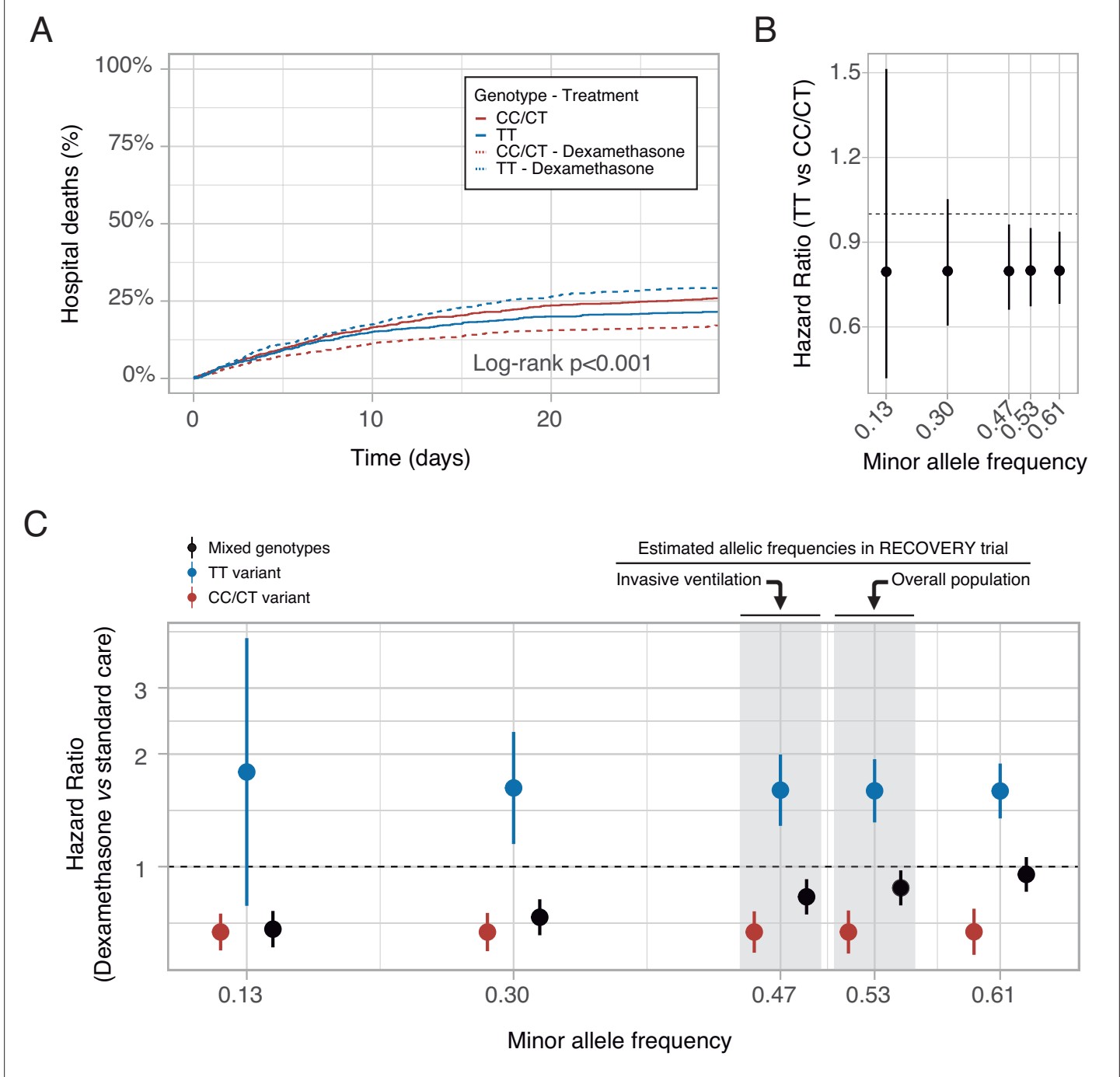

**Figure 7.** In-silico clinical trials. (**A**) Mortality curves modeled using data from the RECOVERY clinical trial, for each rs1990760 variant and treatment allocation, assuming a sample size of 500 patients per group. (**B**) Effect of rs1990760 TT variants (expressed as hazard ratio, HR) in simulated clinical trials including 6000 patients from populations with different minor allele frequencies, assigned to standard care. (**C**) Effect of dexamethasone therapy according to rs1990760 variants and different minor allele frequencies in simulated clinical trials including 6000 patients. Minor allele frequencies were estimated according to reported race in several subsets of patients from the RECOVERY trial: European population (0.61), overall population (0.53), mechanically ventilated patients (0.47), and African American/Asian population (0.13). An intermediate value of 0.30 was added for illustrative purposes. Points and error bars represent the estimated HR and the corresponding 95% confidence interval.

The online version of this article includes the following figure supplement(s) for figure 7:

**Figure supplement 1.** Simulated survival curves corresponding to in-silico clinical trials testing dexamethasone in 6000 COVID-19 patients from populations with different allelic frequencies.

effect of rs1990760 on MDA5 function is related to modifications on the molecular architecture (*Berke and Modis, 2012*; *Wu et al., 2013*) and has been reported by several previous studies (*Looney et al., 2015*; *Nejentsev et al., 2009*).

Collectively, our findings raise the hypothesis that the TT variant could be related to an attenuated pro-inflammatory response. In line with this observation, a TT variant has been associated with lower pro-inflammatory cytokine levels in patients with lupus (*Robinson et al., 2011*; *Zhang et al., 2018*, p. 1) and in experimental viral infections (*Domsgen et al., 2016*).

Development of tolerance to viral diseases can be considered a major evolutionary adaptative response (*Schneider and Ayres, 2008*), and inhibition of pro-inflammatory responses has been proposed to improve the outcome of severe COVID-19 patients. Notably, pangolins, an intermediate host of coronaviruses (*Xiao et al., 2020*), lack functional *IFIH1*. It has been suggested that this deficiency reduces the inflammatory response to coronavirus infections (*Fischer et al., 2020*). IL-6 blockade (*REMAP-CAP Investigators et al., 2021*) or steroids (*WHO Rapid Evidence Appraisal for COVID-19 Therapies (REACT) Working Group et al., 2020*) are the only treatments that have improved the outcome of critically ill COVID-19 patients. However, mortality rates are still around 27–32% (*REMAP-CAP Investigators et al., 2021*; *WHO Rapid Evidence Appraisal for COVID-19 Therapies (REACT) Working Group et al., 2020*), so new therapeutic approaches are warranted.

According to our results, the impact of the treatment with steroids along a given population may depend on intrinsic individual characteristics, allowing a personalized approach based on a specific genomic biomarker. In our study sample, patients with the *IFIH1* rs1990760 TT variant constitute a population with a better prognosis, in whom treatment with dexamethasone may be reconsidered as it was associated to higher mortality rates. Indeed, as shown by the in-silico trials, any population enriched for patients with CC/CT variants in rs1990760 will show higher mortality rates and a better response to steroids, due to the low proportion of TT variants. Dexamethasone in patients receiving mechanical ventilation at inclusion in the RECOVERY trial had an HR of 0.64, compared to an HR of 0.83 in the whole population. Of note, the proportion of Black/Asian patients (with a low frequency of a T allele) in the mechanically ventilated group was 29%, but only 18% in the whole study population. To date, no other genomic markers of personalized therapies in critically ill patients have been identified.

The absence of differences in viral clearance and the late increase in IL-6 in these patients suggest that this worse outcome is more related to the disruption of ongoing immunoregulatory mechanisms than to antiviral responses. Our ex-vivo experiments do not fully elucidate the molecular mechanisms behind the interaction between TT variants and steroids, as the effects of *FOXO3* upregulation and *IL6* expression may be variable (*Joseph et al., 2016*), but clearly illustrate that this specific combination may disrupt the ongoing self-regulation of inflammation. Moreover, steroid-induced changes in gene expression are qualitatively different in patients of each variant.

Our findings point toward the therapeutic potential of MDA5 modulation in COVID-19, either induced by steroids or targeted by other drugs. Amelioration of MDA5-dependent RNA sensing could avoid an exacerbated inflammatory response without impairing viral clearance. In this sense, it has been described that the interferon response triggered by MDA5 is unable to control viral replication (*Rebendenne et al., 2021*). However, genetic ablation of *IFIH1* results in increased viral loads and decreased cytokine production (*Yin et al., 2021*), so this approach must be viewed with caution.

It is unclear if these findings can be translated to other viral diseases. Steroids may decrease mortality in an unselected ARDS population (*Villar et al., 2020*), but the role of underlying genotypes has not been addressed. An experimental model of Coxsackie virus infection revealed lower *IFIH1* and *CXCL10* expression in cells with the TT variant (*Domsgen et al., 2016*), with no relevant differences in viral clearance. However, total absence of MDA5 results in impaired clearance of West Nile virus (*Errett et al., 2013*).

Our work has several limitations. First, the results must be validated in an independent cohort. Our in-silico simulations reinforce the external validity of our findings, but a pharmacogenomic analysis of patients included in clinical trials is warranted for confirmation. Second, steroid treatment was not randomized, so we cannot discard other underlying factors responsible for the observed differences. Although there could be confounding by indication, there were no baseline differences among groups suggesting a higher severity in steroid-treated patients. Moreover, there are significant differences between genotypes irrespective of the treatment received. In addition, in-silico and ex-vivo

experiments are congruent with the observed clinical results, supporting the differential effects of steroid therapy in *IFIH1* rs1990760 variants. Third, the favorable outcome of patients with a TT allele without steroids is based on a small sample size. As steroids are now the standard of care for severe COVID-19, this sample size cannot be increased outside a hypothetical clinical trial focused of personalized steroid therapy according to *IFIH1* rs1990760 variants. However, similar reduced numbers have served to identify other variants in the immune response (*Bastard, 2020*), and the related finding of increased mortality in this genotype after steroid therapy compared to all other variants is supported by a larger sample. In addition, our sample is representative of European population with a limited racial diversity that may have influenced our results, as the in-silico analyses further suggest. Finally, other studies *Pairo-Castineira et al., 2020*; *Ellinghaus et al., 2020* have focused on the genetic variants linked to an increased risk of severe COVID-19, compared to non-infected populations. However, no genetic markers have been associated to mortality in cohorts of infected patients. According to our data, rs1990760 is linked to the outcome, but no inferences on susceptibility to severe COVID-19 can be extracted.

In summary, we have identified a genetic variant of *IFIH1* that results in an ameliorated inflammatory response after severe SARS-CoV-2 infection. Patients with the rs1990760 TT genotype show a good outcome. However, this adaptative response was not observed in patients with a TT variant and treated with steroids. These findings suggest that the systemic response to severe COVID-19 is regulated by genetic factors that modulate the response to the infection and the prescribed therapy and, ultimately, may impact the outcome.

| Genotype | Patients | Standard care | Dexamethasone |
|---|---|---|---|
| | 4689 | 849/3139 | 401/1550 |
| CC/CT | 2954 | a/1978 | b/976 |
| TT | 1735 | (849 a)/1161 | (401-b)/574 |

## Acknowledgements

The authors want to thank all the personnel at the participating ICUs and laboratories for their support during the development of the study. In addition, the authors thank Salvador Villalgordo, Silvia Viñas, Salvador Balboa, Rodrigo Albillos, Cecilia del Busto, Emilio García-Prieto, Jose Antonio Gonzalo, Diego Parra, Lisardo Iglesias, José Alonso, Sérida Dominguez, and Alfredo González their additional efforts for sample collection.

## Additional information

### Funding

| Funder | Grant reference number | Author |
|---|---|---|
| Instituto de Salud Carlos III | PI19/00184 | Carlos Lopez-Larrea |
| Instituto de Salud Carlos III | PI20/01360 | Guillermo M Albaiceta |
| Centro de investigación biomédica en red (CIBER)-Enfermedades respiratorias | CB17/06/00021 | Guillermo M Albaiceta |
| Fundació la Marató de TV3 | grant 413/C/2021 | Laura Amado-Rodríguez |
| Instituto de Salud Carlos III | CM20/00083 | Raquel Rodriguez-Garcia |

The funders had no role in study design, data collection and interpretation, or the decision to submit the work for publication.

### Author contributions

Laura Amado-Rodríguez, Conceptualization, Data curation, Formal analysis, Funding acquisition, Investigation, Methodology, Project administration, Supervision, Validation, Visualization, Writing

– original draft, Writing – review and editing; Estefania Salgado del Riego, Investigation, Resources, Writing – review and editing; Juan Gomez de Ona, Data curation, Investigation; Inés López Alonso, Conceptualization, Data curation, Formal analysis, Investigation, Supervision, Validation, Writing – review and editing; Helena Gil-Pena, Data curation, Investigation, Supervision, Validation; Cecilia López-Martínez, Data curation, Formal analysis, Investigation, Methodology, Software, Visualization, Writing – review and editing; Paula Martín-Vicente, Data curation, Formal analysis, Investigation, Writing – review and editing; Antonio Lopez-Vazquez, Raquel Rodriguez-Garcia, Marta Elena alvarez-Arguelles, Juan Mayordomo-Colunga, Jose Ramon Vidal-Castineira, Margarita Fernandez, Loreto Criado, Victoria Salvadores, Belen Prieto, Alejandra Fernandez-Fernandez, Investigation, Writing – review and editing; Adrian Gonzalez Lopez, Data curation, Formal analysis, Investigation, Software, Validation, Visualization, Writing – review and editing; Elias Cuesta-Llavona, Data curation, Investigation, Writing – review and editing; Jose Antonio Boga, Investigation, Methodology, Writing – review and editing; Irene Crespo, Investigation, Validation, Writing – review and editing; Francisco Jose Jimeno-Demuth, Formal analysis, Investigation, Software, Writing – review and editing; Lluis Blanch, Methodology, Supervision, Validation, Writing – review and editing; Carlos Lopez-Larrea, Conceptualization, Funding acquisition, Investigation, Methodology, Supervision, Validation, Writing – review and editing; Eliecer Coto, Conceptualization, Data curation, Funding acquisition, Investigation, Methodology, Supervision, Validation, Writing – review and editing; Guillermo M Albaiceta, Conceptualization, Data curation, Formal analysis, Funding acquisition, Investigation, Methodology, Project administration, Resources, Software, Supervision, Validation, Visualization, Writing – original draft, Writing – review and editing

#### Author ORCIDs
Laura Amado-Rodríguez (iD) http://orcid.org/0000-0002-8793-0213
Juan Mayordomo-Colunga (iD) http://orcid.org/0000-0003-0997-4410
Guillermo M Albaiceta (iD) http://orcid.org/0000-0002-9276-3253

#### Ethics
Human subjects: This single-center prospective, observational study was approved (ref. 2020/188) by the Clinical Research Ethics Committee of Principado de Asturias (Spain). Informed consent was obtained from each participant or next of kin.

#### Decision letter and Author response
Decision letter https://doi.org/10.7554/eLife.73012.sa1
Author response https://doi.org/10.7554/eLife.73012.sa2

## Additional files

#### Supplementary files
• Supplementary file 1. Supplementary Tables. 1a. Primers and probes used to detect SARS-CoV-2. 1b: Primers used for qPCR of human genes. 1c: Clinical characteristics of patients according to *IFIH1* rs1990760 genotypes and dexamethasone therapy. Values are shown as absolute count or median (interquartile range). PBW: Predicted body weight. NIV: Non-invasive ventilation. PEEP: Positive End-Expiratory Pressure. *p-values calculated for proportion over the number of intubated patients.

• Supplementary file 2. Results of differential expression analysis between rs1990760 variants.

• Supplementary file 3. Changes in gene expression after steroid treatment in rs1990760 variants.

• Transparent reporting form

• Reporting standard 1. STROBE check list for cohort studies.

#### Data availability
RNA-seq data has been deposited in Gene Expression Omnibus (accession numbers GSE168400 and GSE177025). Code for in-silico clinical trials is available at https://github.com/Crit-Lab/IFIH1_simulation (copy archived at swh:1:rev:343c92a4ce58b831df7b289bf40b539db5298d0a). Source data files for figures 3, 4 and 5 have been provided. Patient-derived data related to this article are not publicly available out of respect for the privacy of the patients involved. De-identified clinical data will be shared with researchers with any proposal after review and approval by the local Ethics Committee and a signed data access agreement. All requests should be sent to the corresponding author (gma@crit-lab.org).

The following dataset was generated:

| Author(s) | Year | Dataset title | Dataset URL | Database and Identifier |
|---|---|---|---|---|
| Albaiceta GM, López-Alonso I, López-Martínez C, Amado-Rodríguez L | 2021 | Peripheral blood gene expression according to IFIH1 rs1990760 variants in critically-ill COVID-19 patients | https://www.ncbi.nlm.nih.gov/geo/query/acc.cgi?acc=GSE168400 | NCBI Gene Expression Omnibus, GSE168400 |
| Albaiceta GM, López-Alonso I, López-Martínez C, Amado-Rodríguez L | 2021 | Effects of steroids on peripheral blood gene expression in COVID-19 patients according to rs1990760 variants | https://www.ncbi.nlm.nih.gov/geo/query/acc.cgi?acc=GSE177025 | NCBI Gene Expression Omnibus, GSE177025 |

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

## Appendix 1

### Hazard ratio estimation and in-silico clinical trials

Estimation of hazard ratios from the RECOVERY trial

Our findings raise the hypothesis that steroid therapy will have a larger effect in populations with a low proportion of individuals with a TT genotype. Populations with black/Asian ancestry have an allelic frequency of the T allele of 0.13 (COVID-19 GWAS Results. https://grasp.nhlbi.nih.gov/Covid19GWASResults.aspx). Therefore, the frequency of a TT genotype is 0.017, implying that the majority of the population has a CC/CT variant. Using data from the RECOVERY trial, we assume that the effects of steroids in this population correspond to the effect on patients with a CC/CT variant. In this population, steroids significantly decreased mortality, with a RR of 0.7 (0.51–0.95). Therefore, we assumed that treatment with steroids in patients with these non-TT variants decreases mortality with a RR of 0.7.

White populations have an allelic frequency of the T allele in rs1990760 of 0.61 (COVID-19 GWAS Results. https://grasp.nhlbi.nih.gov/Covid19GWASResults.aspx). Therefore, the proportion of patients with a TT variant is 0.37. GWAS data released by the UK biobank report a coefficient for hospital death in patients with a T allele (compared to positive survivors) of 0.03 (COVID-19 GWAS Results. https://grasp.nhlbi.nih.gov/downloads/COVID19GWAS/05092021/top_UKBB_death_ALLpstv_050921.txt.gz), so the RR for a TT variant (irrespective of the therapy) is $e^{2 \cdot 0.03}$=1.06. Translating these findings to white patients included in the RECOVERY trial yields the following mortality table (assuming that randomization was independent of the rs1990760 variant):

where a and b are the number of deaths in patients with a CC/CT variant assigned to standard care or dexamethasone, respectively.

Considering that the previously calculated RR of steroids in CC/CT population is 0.7, we can estimate:

$$\frac{b}{976} = 0.7 \frac{a}{1978}$$

$$b = 0.3454a$$

And assuming a global RR of a TT genotype of 1.06 (irrespective of therapy), we can estimate:

$$\frac{\frac{(849-a)+(401-b)}{1735}}{\frac{a+b}{2954}} = 1.06$$

Solving these two equations yields the following values:

$$a = 573$$

$$b = 198$$

Therefore, distribution of mortality according to therapy and rs1990760 variants was estimated as:

| Genotype | Patients | Standard care | Dexamethasone |
|---|---|---|---|
|  | 4689 | 849/3139 | 401/1550 |
| CC/CT | 2954 | 573/1978 | 198/976 |
| TT | 1735 | 276/1161 | 203/574 |

From these values, the RR of a TT genotype compared to CC/CT variants in patients not receiving steroids is:

$$RR_{\left(TT vs \frac{CC}{CT}\right) Nosteroids} = \frac{\frac{276}{1161}}{\frac{573}{1978}} = \frac{0.238}{0.290} = 0.821$$

And the RR related to steroid therapy compared to standard care in patients with a TT variant is:

$$RR_{(Dexamethasone vs std.care) TTvariant} = \frac{\frac{203}{574}}{\frac{276}{1161}} = \frac{0.354}{0.238} = 1.487$$

## In-silico clinical trials

The previously estimated RRs were used to simulate mortality curves and perform in-silico clinical trials. Mortality was modeled using an asymptotic regression curve with a cumulative distribution function:

$$F(x) = a - ae^{-ct}$$

where a is the upper asymptote, c is a curvature parameter, e is the base of natural logarithms and t is time. Assuming that final mortality (this is, the upper asymptote) is 10% higher than the observed mortality at day 28 (as in the survival curves of our cohort), we obtain that the curvature parameter is 0.08564. This parameter is constant for all curves, so the differences among groups depend only on the upper asymptote. This model is appropriate to fit acute conditions in which most of the deaths occur in the short term, but a substantial proportion of patients survive the disease. The code for these simulations can be found at https://github.com/Crit-Lab/IFIH1_simulation (copy archived at swh:1:rev:343c92a4ce58b831df7b289bf40b539db5298d0a, *Crit-Lab, 2022*).

Different clinical trials including patients randomized into four groups depending on the rs1990760 variant (CC/CT or TT) and prescription of dexamethasone were simulated. Randomization to dexamethasone or placebo was 1:1. Minor allele frequencies from 0.13 to 0.61 (corresponding to populational distributions) and two baseline mortality rates (by adjusting the mortality of patients with a CC/CT variant not treated with steroids to 25.7% and 41.4%, corresponding to mortality rates of the overall RECOVERY trial and those receiving mechanical ventilation respectively) were tested. Each set of conditions was repeated 1000 times and the average RR was calculated. All the simulations were done using the Mediana package for R (available at https://CRAN.R-project.org/package=Mediana).

