## [Editor Report]

Authors present an interesting approach of COVID-19 multiorgan failure and loss of homeostasis attributed to modified adaptive immune response due to presence/absence of MDA5 polymorphism(rs1990760). Data presented by authors are very interesting representing novel scientific work that adds new and potentially pivotal information to existing knowledge. Key data presented strongly support an individualized approach during COVID-19 pandemic that could eventually alter therapeutic algorithm regarding dexamethasone administration implemented for these patients.

---

## [Decision Letter]

**Decision letter after peer review:**

Thank you for submitting your article "IFIH1 rs1990760 variants, systemic inflammation and outcome in critically-ill COVID-19 patients in an observational translational study" for consideration by *eLife*. Your article has been reviewed by 2 peer reviewers, and the evaluation has been overseen by a Reviewing Editor and Jos Van der Meer as the Senior Editor. The reviewers have opted to remain anonymous.

Essential revisions:

– Methods: poly-I:C/Lyovec, Invivogen is described by manufacturer as both RIG-I and MDA-5 agonist, but RIG-1 and MDA5 represent distinct signaling pathways ultimately regulating type I IFN gene expression. Authors used this MDA-5 ligand as part of their ex vivo experiment of mimicking SARS-Cov-2 infection interpreting results solely by MDA-5 triggering. Further clarification at this point is needed as this could represent a possible confounder factor.

– Results: Authors state there were no significant differences in the main clinical characteristics among the resulting groups, but P/F ratio was lower at TT group. Also, CC/CT groups not received steroids were not studied separately (27 put of 35), how authors propose to investigate patient carrying at least on T allele needs further clarification. Overall dexamethasone induced IL-6 secretion in patients carrying TT-variants in line with ex-vivo experiments suggesting that dexamethasone administration acted as an anti-inflammatory factor for CC/CT patients but promoted a pro-inflammatory profile in TT group which was subsequently linked with ominous outcome.

– Could authors provide a plausible explanation why serum concentration of interferons was not affected by the presence of TT variant?

– Table 1: All data are shown as median (interquartile range). Could the authors confirm that all variables, including age, BMI, and others in Table 1 are non-parametric? Also, disease severity scores such as SOFA or APACHE are missing. Differences in disease severity could be an important confounder for the presented results. For instance, is there disease severity difference between the CC/CT and TT groups, which may explain the difference in serum biomarkers? Do the steroid / no-steroid groups have different disease severity (supplemental table)?

– Figure 2: Figure 2A shows that IFIH1 is downregulated in (most of the) patients with the rs1990760 TT variant, although this single-gene direct comparison does not involve multiple testing correction, of course. Importantly, IFIH1 is not among the significant DEGs after correction for multiple testing. This appears to be indicating that IFIH1 expression is not very different between the variant groups. If so, how much of the differences between the CC/CT and TT groups can realistically be attributed to IFIH1-mediated mechanisms?

– Figure 3: "Patients with the TT genotype have lower inferred percentages of classic CD14+ monocytes and circulating plasma cells, and an increase in hematopoietic precursors, myeloid dendritic cells, M2 macrophages and CD56dim NK cells (Figure 3)."

This is not correct, CD14 monocytes and plasma cells were not significantly different (P>0.05) according to Figure 3. The same holds for M2 macrophages and the NK cell subset. Given the already suboptimal method of deconvolution from bulk RNA data (the obtained correlation coefficient of deconvoluted percentages and the measured lymphocyte percentages was 0.61, which does not provide much confidence on the robustness of this approach), these non-significant differences should not be interpreted as significant.

– Figure 4: It is not clear if the lower limit of the Y-axis on the boxplots is zero. In any case, it appears that the statistics in the boxplots do not take into account the lowest values (for example IFN-γ): the median and IQR seem unaffected by the zero values. This could be a consequence of replacing values below the LLOD with zero, instead of imputation or replacement with half of the LLOD. Could the authors please clarify?

– The differential dexamethasone effect on clinical parameters and outcomes between the variant groups is very interesting, and well examined in this study. As stated before, disease severity scores should be part of the descriptive table of these groups.

---

## [Author Response]

Essential revisions:– Methods: poly-I:C/Lyovec, Invivogen is described by manufacturer as both RIG-I and MDA-5 agonist, but RIG-1 and MDA5 represent distinct signaling pathways ultimately regulating type I IFN gene expression. Authors used this MDA-5 ligand as part of their ex vivo experiment of mimicking SARS-Cov-2 infection interpreting results solely by MDA-5 triggering. Further clarification at this point is needed as this could represent a possible confounder factor.

Thanks for this comment, we agree that it deserves clarification, given that cytoplasmic RNA sensors include both RIG-I and MDA-5, as pointed by the reviewer. For our ex vivo experiments, high molecular weight (HMW) Poly(I:C) was employed. This molecule predominantly stimulates MDA5 rather than RIG-I. RIG-I detects dsRNAs without a 5’-triphosphate end and rather short than long dsRNAs. Conversely, MDA5 selectively recognizes long dsRNAs which explains why HMW Poly(I:C), a long synthetic analog of dsRNA (https://www.invivogen.com/polyic-hmw-lyovec), behaves preferentially as a MDA5 ligand. We have added a reference (Kato et al., 2008) describing this specificity.

– Results: Authors state there were no significant differences in the main clinical characteristics among the resulting groups, but P/F ratio was lower at TT group.

The reviewer is absolutely right. A sentence in the first paragraph of the Results section states that the only clinical variable significantly different between groups was P/F ratio, which could be speculated as a marker of increased severity although the difference in absolute values is not clinically relevant.

Also, CC/CT groups not received steroids were not studied separately (27 put of 35), how authors propose to investigate patient carrying at least on T allele needs further clarification. Overall dexamethasone induced IL-6 secretion in patients carrying TT-variants in line with ex-vivo experiments suggesting that dexamethasone administration acted as an anti-inflammatory factor for CC/CT patients but promoted a pro-inflammatory profile in TT group which was subsequently linked with ominous outcome.

As the reviewer points out, CC anc CT variants were grouped in our study. The reason behind grouping these two groups is that previous literature on rs1990760 variants suggests that the TT variant is associated to a different inflammatory profile. Therefore, the CC and CT variants were grouped to increase statistical power. This rationale has been added to the manuscript (page 8). In addition, we performed an additional analysis comparing these two genotypes. There were no significant differences in any of the measured cytokines between these two groups, further confirming this similar behavior. This has been added to the manuscript (page 14 and supplementary figure 5).

– Could authors provide a plausible explanation why serum concentration of interferons was not affected by the presence of TT variant?

Thank you very much for this interesting comment. Serum IFNs levels were mostly below the inferior detection threshold. This result is similar to the one reported by Galani et al., (PMID 33277638), where IFN-λ and type I IFN production were both diminished and delayed in moderate-to-severe COVID-19 patients followed-up during hospitalization. In our study, IFNs levels were only measured on day 1 of ICU admission, precluding any conclusions regarding time-dependent changes on these mediators’ levels during, for instance, subacute or long-term infection. Besides, recognition of viral RNA by cytosolic receptors leading to the ultimate induction of expression of types I and II IFNs as part of the activation of innate antiviral signaling cascades, may not only relay on MDA5 receptor but also on other retinoid acid-inducible gene (RIG)-I-like receptors, or on the endosomal Toll-like receptor 3 (TLR3) (20303872). These other recognition actors could have a predominant role in those patients lacking normal MDA5 expression during the acute SARS-CoV-2 infection. These clarifications have been included in the Discussion section (page 18-19).

– Table 1: All data are shown as median (interquartile range). Could the authors confirm that all variables, including age, BMI, and others in Table 1 are non-parametric? Also, disease severity scores such as SOFA or APACHE are missing. Differences in disease severity could be an important confounder for the presented results. For instance, is there disease severity difference between the CC/CT and TT groups, which may explain the difference in serum biomarkers? Do the steroid / no-steroid groups have different disease severity (supplemental table)?

Thank you for this observation. We confirm that non-parametric statistics were applied whenever distribution’s parameters for a given variable remained unspecified or not normal.

We agree that severity assessment is essential to understand the results. We have computed APACHE-II scores in the sample, and added the results to Table 1 and Supplementary Table 1. There are no significant differences among groups in this severity index.

– Figure 2: Figure 2A shows that IFIH1 is downregulated in (most of the) patients with the rs1990760 TT variant, although this single-gene direct comparison does not involve multiple testing correction, of course. Importantly, IFIH1 is not among the significant DEGs after correction for multiple testing. This appears to be indicating that IFIH1 expression is not very different between the variant groups. If so, how much of the differences between the CC/CT and TT groups can realistically be attributed to IFIH1-mediated mechanisms?

Thank you for this comment. There is growing evidence reporting the effect of rs1990760, along with other IFIH1 gene variants, on MDA5 function (PMIDs 26385483, 25579795, 19264985, 24530055) rather than on protein expression. RIG-1-like receptor family shares two common domains: a C-terminal domain and an adjacent helicase domain. Both together conform the dsRNA binding pocket. MDA5, a member of this family, has also a third domain that interacts with the adapter molecule mitochondrial antiviral-signaling protein (MAVS) and corresponds to the N-terminal caspase recruitment domain. These few domain architectures permit the recognition of extremely different ligands and the consequent acquisition of biological adaptative functions. The molecular architecture of IFIH1-encoded MDA5 protein, rather than IFIH1 expression levels alone, may be modified according to genetic variants affecting IFIH1 gene sequence. For instance, rs1990760 resides within the C-terminal domain. Deletion of the MDA5 C-terminal loop structure containing rs1990760 SNP did not affect dsRNA binding but increased ATPase activity and reduced signaling activity (PMID 23273991). ATPase activity regulates assembly and maintenance of the helical filaments formed by MDA5 along dsRNA. This C-terminal loop structure is critical to ATPase activity and to intermolecular stability when MDA5 forms these helical filaments along dsRNA (PMID 22314235). Besides, rs1990760 is a non-synonymous mutation occurring within the binding site of transcription factors that may be related to altered function of MDA5 protein, even when similar expression levels of IFIH1 may follow different genetic variants.

Thus, although expression levels may be similar, full activation of an anti-viral response triggering specific signaling intermediates and transcription factors, may vary between genotypes due to differences in MDA5 molecular architecture and function. A paragraph clarifying this point has been added to the discussion.

– Figure 3: "Patients with the TT genotype have lower inferred percentages of classic CD14+ monocytes and circulating plasma cells, and an increase in hematopoietic precursors, myeloid dendritic cells, M2 macrophages and CD56dim NK cells (Figure 3)."This is not correct, CD14 monocytes and plasma cells were not significantly different (P>0.05) according to Figure 3. The same holds for M2 macrophages and the NK cell subset. Given the already suboptimal method of deconvolution from bulk RNA data (the obtained correlation coefficient of deconvoluted percentages and the measured lymphocyte percentages was 0.61, which does not provide much confidence on the robustness of this approach), these non-significant differences should not be interpreted as significant.

The reviewer is right. The manuscript has been corrected accordingly.

– Figure 4: It is not clear if the lower limit of the Y-axis on the boxplots is zero. In any case, it appears that the statistics in the boxplots do not take into account the lowest values (for example IFN-γ): the median and IQR seem unaffected by the zero values. This could be a consequence of replacing values below the LLOD with zero, instead of imputation or replacement with half of the LLOD. Could the authors please clarify?

Thank you very much for this comment. As the reviewer points out, values below the LLOD were replaced by zeroes, and plotting these values in a log scale gives the artifact and does not take the values into account to draw the boxplot. We have followed the reviewer’s suggestion, and values below the LLOD were substituted by half of the LLOD, and the plots corrected. As the zeros were taken into account for statistical comparison, and the Wilcoxon test is based in ranks, not in mean values, the p values of the panes is the same. The methods section dealing with replacement of values below the LLOD has been modified accordingly.

Additionally, after reviewing these data, we noticed a mistake in IFN-λ units that has been corrected (reported values are actually 100 times lower than in the first version of the manuscript). Figures and manuscript have been corrected accordingly.

– The differential dexamethasone effect on clinical parameters and outcomes between the variant groups is very interesting, and well examined in this study. As stated before, disease severity scores should be part of the descriptive table of these groups.

Thank you again for the suggestion. APACHE-II scores have been added to tables. All groups had similar scores.